

# Natural products as promising therapeutics for fine particulate matter–induced skin damage: a review of pre-clinical studies on skin inflammation and barrier dysfunction

Saowanee Jeayeng[1,2,*], Jaturon Kwanthongdee[3,*], Ratima Jittreeprasert[3], Kankanich Runganantchai[3], Kalayaporn Naksavasdi[3], Rosarin Rirkkrai[3], Varisara Wongcharoenthavorn[3], Wiriya Mahikul[3] and Anyamanee Chatsirisupachai[3]

[1] Department of Medical Science, School of Medicine, Walailak University, Nakhon Si Thammarat, Thailand
[2] Center of Excellence in Tropical Pathobiology, Walailak University, Nakhon Si Thammarat, Thailand
[3] Princess Srisavangavadhana Faculty of Medicine, Chulabhorn Royal Academy, Bangkok, Thailand
[*] These authors contributed equally to this work.

Corresponding author
Anyamanee Chatsirisupachai,
anyamanee.cha@cra.ac.th

## ABSTRACT

**Background**. Particulate matter less than 2.5 $\mu$m (PM2.5) is a significant air pollutant and is linked to an increased risk of health conditions, including skin diseases. The skin, as the first barrier and the largest organ, is primarily damaged by PM2.5 through different pathways. Several studies have shown that PM2.5 upregulates inflammatory responses through the excessive production of reactive oxygen species (ROS) and several inflammatory cytokines, leading to PM2.5-induced skin damage. The ROS/mitogen-activated protein kinase (MAPK) and Cyclooxygenase-2-Prostaglandin E2 (COX2/PGE2) inflammatory pathways are activated by free radical scavenging and phase II detoxification. Natural products have been suggested as therapeutic agents for mitigating PM2.5-induced skin damage.

**Objectives**. We elaborate on the mechanisms of action of natural products and their functions as protectants against environmental skin diseases. This review highlights the optimal doses of natural products for clinical study, which may benefit dermatologists, molecular biologists, clinicians, and healthcare professionals in preventive and alternative medicine.

**Methodology**. The available scientific literature published between 1999 and 2024 was searched using PubMed and Google Scholar. Multiple keywords related to the topic were used. Only 41 of the screened articles were chosen for this review, as they were the most relevant publications on the topic of the preventive advantages of natural products and specific pathways targeting PM2.5-induced skin injury. All relevant articles meeting the criteria of being original full articles and written in English were included.

**Results**. This review summarized the natural products, including phenolic/polyphenolic compounds and flavonoids, that can act as anti-inflammatory and antioxidant agents by protecting the skin against oxidative stress, inhibiting enzymes that promote free radical formation, enhancing antioxidant enzyme activity, and reducing overall

## INTRODUCTION

Particulate matter less than 2.5 micrometers in size (PM2.5) (*Thangavel, Park & Lee, 2022*), consists of particles composed of inorganic ions, carbonaceous compounds, and mineral dust (*McDuffie et al., 2021*). In recent years, the levels of PM2.5 have increased due to the combustion of biomass (*Suriyawong et al., 2023*) and fuels (*Vohra et al., 2021*). PM2.5 has been associated with an increased risk of various diseases, including respiratory damage (*Huang et al., 2021*), cardiovascular disease (*Krittanawong et al., 2023*), chronic kidney disease (*Xu et al., 2022*), intellectual development disorders (*Chang et al., 2023*; *McGuinn et al., 2020*), and Alzheimer's disease (*Fu et al., 2022*; *Yang et al., 2022*).

As the primary protective barrier against environmental pollutants, the skin is profoundly affected by PM2.5 exposure, which has been implicated in various dermatological conditions, including premature aging, acne, and skin cancer. The harmful effects of PM2.5 are due to its ability to penetrate the skin barrier and trigger oxidative stress, inflammation, and molecular damage, which disrupt skin homeostasis and exacerbate existing dermatologic conditions. One of the most worrying effects of PM2.5 exposure is premature skin aging, which results from the breakdown of collagen and elastin fibers and leads to wrinkling, loss of elasticity, and hyperpigmentation. At the molecular level, PM2.5-induced oxidative stress damages cellular components and activates inflammatory pathways that upregulate proinflammatory cytokines and matrix metalloproteinases (MMPs), which further accelerate extracellular matrix degradation and skin aging (*Abolhasani et al., 2021*; *Diao et al., 2021*; *Paik et al., 2024*).

In addition to premature skin aging, exposure to particulate matter is strongly associated with acne and other inflammatory skin conditions. Studies suggest that PM2.5 stimulates sebaceous gland activity, increases sebum production, and promotes inflammation, resulting in an exacerbation of the severity of acne (*Paik et al., 2024*). Other skin diseases,

such as atopic dermatitis and psoriasis, are also exacerbated by particulate matter, as it disrupts keratinocyte differentiation, impairs epidermal barrier function, and triggers chronic immune activation (*Abolhasani et al., 2021*). The inflammatory response triggered by PM2.5 leads to increased skin sensitivity, erythema, and persistent irritation, making individuals with preexisting skin conditions more susceptible to flare-ups. Aside from inflammation and aging, long-term exposure to PM2.5 has also been linked to an increased risk of skin cancer, including melanoma and squamous cell carcinoma. The cumulative effects of oxidative stress, inflammation, and DNA damage caused by PM2.5 contribute to carcinogenesis, particularly due to the sustained activation of reactive oxygen species (ROS) and proinflammatory mediators (*Abolhasani et al., 2021*). Although the harmful effects of PM2.5 on skin health are well documented, emerging research suggests that protective measures, such as topical antioxidants, barrier-enhancing treatments, and skin care formulations that protect against pollution, can mitigate these effects (*Ferrara et al., 2024*; *Park et al., 2023*). These findings highlight the growing need for effective skin care products and therapeutic interventions to combat environmental skin damage, while also emphasizing the importance of ongoing research into dermatological protection against environmental pollutants, such as PM2.5.

PM2.5 generates ROS that cause oxidative stress, which impairs cell viability, induces DNA damage and lipid peroxidation, and alters protein carbonylation. At the cellular level, this damage in turn triggers endoplasmic reticulum (ER) stress, mitochondrial swelling, autophagy, and cell apoptosis, as demonstrated in both HaCaT cells and mouse skin tissue (*Piao et al., 2018*; *Zhen et al., 2019*). PM2.5 has been extensively studied for its role in disrupting the epidermal skin barrier through AhR- and Th17 cell-mediated inflammatory pathways (*Kim et al., 2023*). Interestingly, several subtypes of polycyclic aromatic hydrocarbons (PAHs), a component of particulate matter, have been identified as potent AhR agonists. The aryl hydrocarbon receptor (AhR) has been reported as a transcription factor in detoxifying xenobiotics and regulating the immune cells (*Lee et al., 2015*). The PAHs present in particulate matter play a role in AhR-mediated skin damage by activating several downstream pathways involved in the upregulation of CYP1A1 and the AhR/Nox2/p47phox pathway. Exposure to PM2.5 causes ERK/p38-dependent signaling as well as NF-κB- and JNK-dependent activation of activator protein 1 (AP-1) transcription factors. In addition, the enhancement of c-Src activity by AhR activation leads to EGFR activation and downstream MAPK signaling (*Diao et al., 2021*).

Previous research has demonstrated that PM2.5 exposure leads to a significant increase in chemokines, including chemokine C-X-C motif ligand 1 (CXCL1) and interleukin 8 (IL-8), which trigger neutrophil chemotaxis and upregulate inflammatory responses in the skin (*Kono et al., 2023*). The development of allergic diseases, such as atopic dermatitis, is associated with PM2.5-induced TNF-α, which contributes to a deficiency of filaggrin (FLG) in the skin and leads to further impairment of the skin barrier function (*Kim et al., 2021*). Exposure of HaCaT keratinocytes to PM2.5 has been shown to modulate the activity of apoptotic proteins, including the upregulation of proapoptotic proteins, such as Bax and caspase 9, along with the downregulation of the antiapoptotic protein Bcl-2 (*Zhu et al., 2022*) and caspase 3 (*Shan et al., 2022*).

Given the skin toxicity of PM2.5 and the heightened inflammatory response it induces, ongoing therapeutic interventions and scientific research are presently focused on mitigating these adverse effects. One promising therapy is the use of natural substances that exhibit anti-inflammatory properties and promote the inhibition of excessive ROS formation (*Bae et al., 2021*; *Liao, Nie & Sun, 2020*). For example, phenolic compounds, flavonoid compounds, nonflavonoid phenolic compounds, and phytosterols, which have known effects on the ROS/MAPK and COX2/PGE2 pathways (*Diao et al., 2021*; *Greenhough et al., 2009*; *Son et al., 2011*), could potentially have protective effects when applied to skin exposed to PM2.5.

Natural products have long been utilized to enhance health, and they have served as a valuable source for developing novel medications. They have demonstrated significant therapeutic potential for treating or preventing global health issues, including metabolic and cardiovascular diseases. Recently, traditional medicine has received renewed focus, as remedies derived naturally from plants have had their safety and efficacy established through both preclinical and clinical studies. Herbal medicines have shown promise in protecting against infections, thereby offering a viable alternative to combat existing and emerging drug-resistant pathogens while avoiding the adverse effects of drug metabolism, such as nausea, stomachache, and unconsciousness (*Rossiter, Fletcher & Wuest, 2017*; *van Wietmarschen et al., 2022*). The main objective of the present review is to provide a comprehensive analysis of the protective effects of natural products against PM2.5-induced skin damage, focusing on their anti-inflammatory, antioxidant, and skin barrier–preserving properties.

## METHODOLOGY

### Search strategy

In this study, our aim was to update and integrate the information contained in publications retrieved from Google Scholar and PubMed, which are the top academic search engines related to the medical field. The search included articles published from 1999 to 2024. The search was performed using the following search terms, together with relevant Medical Subject Headings (MeSH) terms identified for individual databases: ((("particulate matter"[MeSH Terms] OR ("particulate"[All Fields] AND "matter"[All Fields]) OR "particulate matter"[All Fields]) AND ("biological products"[MeSH Terms] OR ("biological"[All Fields] AND "products"[All Fields]) OR "biological products"[All Fields] OR ("natural"[All Fields] AND "products"[All Fields]) OR "natural products"[All Fields]) AND ("skin"[MeSH Terms] OR "skin"[All Fields]) NOT ("clinical trial"[Publication Type] OR "clinical trials as topic"[MeSH Terms] OR "clinical trial"[All Fields])) AND (1999:2024[pdat]). The publications were selected according to specific inclusion and exclusion criteria to identify the most appropriate manuscripts retrieved from these two databases.

### Eligibility criteria

The research question was constructed and developed according to the Population, Intervention, Comparison, Outcome, and Study Design (PICOS) framework. To be

eligible for inclusion in this study, a research article had to meet the following inclusion and exclusion criteria: the study design should be focused on a skin cell line exposed to particulate matter 2.5, rather than other pollutants (*e.g.*, particulate matter 10 μm (PM10), ambient air, air pollutants, or dust particles); the study should be focused on treatments using natural products rather than synthetic chemical products; the study should include preclinical experiments conducted *in vitro* or *in vivo*; the article should be published in English as well as an original full research article. We excluded narrative reviews, experts' opinions, and articles published in other languages.

A total of 78 articles were selected and identified by their appropriate titles as manuscripts contributing to cell analysis. Subsequent screening was based on the contents of the abstract and the presence of keywords such as "natural products," "active compounds," "phytochemicals," "antioxidants," "skin," "keratinocytes," "fibroblast," and "particulate matter less than 2.5 or PM2.5." Approximately 37 of the 78 articles were excluded based on the inclusion and exclusion criteria established for this review. Overall, references were extracted from 41 articles for this review.

## Importance of natural products as protectants against PM2.5-induced skin damage

Many recent studies have reported the use of natural products to protect the skin from particulate matter. Natural products and their structural counterparts have traditionally made a significant contribution to pharmacotherapy and have played a crucial role in drug development. Previously, *Centella asiatica*, a medicinal plant widely used in Southeast Asia, has long served as an alternative medicine for atopic dermatitis, a skin condition characterized by an abnormal immune response triggered by environmental factors, such as air pollution (*Lee et al., 2020*). It is the secondary plant metabolites present in medicinal plants that impart their medicinal properties. For example, phenolic compounds (PCs) are naturally occurring antioxidants whose chemical structure contains hydroxyl groups that may give these metabolites the ability to scavenge free radicals, such as ROS (*Platzer et al., 2022*). Similarly, the presence of hydroxyl groups in flavonoid compounds can impart protective antioxidant, anti-inflammatory, and anti-aging effects when flavonoids are applied to the skin (*Diao et al., 2021*). Flavonoids are also known for their scavenging activity, which can neutralize free radicals (*Panche, Diwan & Chandra, 2016*). Phytosterols (PS), another class of secondary metabolites that can be isolated from plants, also have anti-inflammatory, antioxidant, antidiabetic, and chemopreventive effects (*Salehi et al., 2021*). The PS function through the NF-κB and MAPK signaling pathways to suppress the expressions of COX-2, PGE2, and proinflammatory cytokines, thereby reducing the levels of inflammatory mediators (*Diao et al., 2021*).

## Protective effects of natural products based on the classification of their phytochemical structures

Phenolic compounds, which are present in a variety of foods, including fruits, vegetables, and beverages, which can exhibit antioxidant, anti-inflammatory, anti-allergic, and anticancer properties. Due to their cardioprotective and other health-promoting benefits, they are widely incorporated into functional foods (*Alara, Abdurahman & Ukaegbu, 2021*).

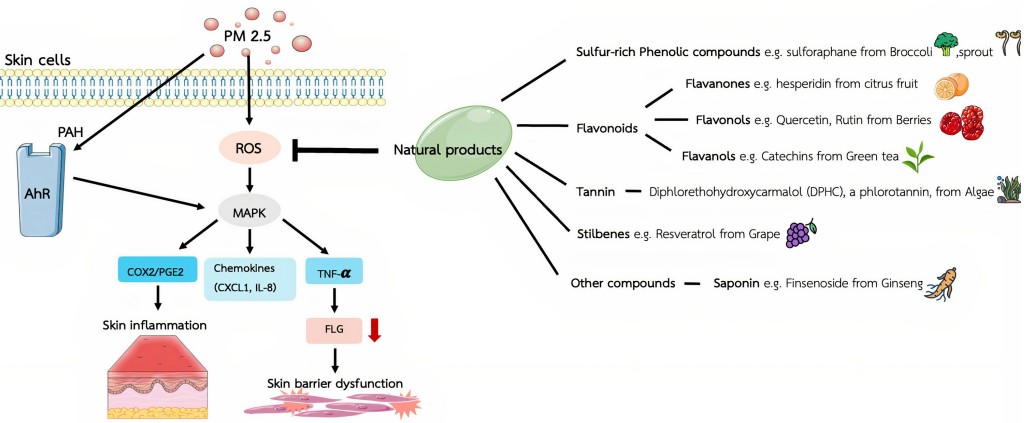

**Figure 1** **The mechanistic findings obtained from *in vitro* studies on the protective effects of herbal chemicals against fine particulate matter–induced skin damage.** Polycyclic aromatic hydrocarbons (PAHs), a group of fused ring aromatic compounds, are recognized as a major toxic component of the environmental pollutant PM 2.5, which contribute to the aryl hydrocarbon receptor (AhR)-mediated skin damage by triggering several downstream pathways. Particulate matter 2.5 (PM2.5) upregulates the inflammatory responses by enhancing excessive reactive oxygen species (ROS) and several inflammatory cytokines, leading to PM2.5-induced skin damage including mitogen-activated protein kinase (MAPK), Cyclooxygenase-2-Prostaglandin E2 (COX2/PGE2) inflammatory pathways, chemokines secretion of chemokine C-X-C motif ligand 1 (CXCL1) and interleukin-8 (IL-8), and tumor necrosis factor-α (TNF-α)/filaggrin(filament aggregating protein; FLG), which responsible for the skin inflammation and barrier dysfunction. Figure was drawn in part using images from Servier Medical Art. Servier Medical Art by Servier is licensed under a Creative Commons Attribution 4.0 Unported License (https://creativecommons.org/licenses/by/4.0/).

One class of phenolic compounds—polyphenols—is a group of chemical molecules that contain multiple hydroxyl structural units on aromatic rings that impart strong antioxidant properties. Differences in the molecular backbone structure of polyphenols have led to their general categorization into phenolic acids, flavonoids, stilbenes, and lignans. Flavonoids, as the largest group of polyphenols, are divided into seven subgroups according to their chemical and biological properties: flavanols (yellow), flavanones (purple), flavones (light blue), flavonols (red), isoflavones (dark blue), anthocyanins (green), and others (orange) (*Wang et al., 2022a*; *Zhang et al., 2022*). The flavonoid colors are useful for visually distinguishing the different flavonoid subgroups based on their structural characteristics. Other types of compounds, saponins are a type of triterpene glycoside with a bitter taste. Structurally, saponins are glycosides; that is, sugars are bound to one or more organic molecules. In this review, the effects of bioactive compounds extracted from natural products are presented based on plant organic chemicals (Fig. 1), and the compounds showing protective action against particulate matter–induced skin damage are summarized in Table 1.

## Phenolic compounds—sulforaphane from broccoli and sprouts

Sulforaphane is a polyphenolic compound found in cruciferous vegetables, such as broccoli. It is produced through the enzymatic conversion of glucoraphanin by myrosinase, an enzyme presents in plants. Sulforaphane is metabolized *via* the mercapturic acid

Jeayeng et al. (2025), *PeerJ*, DOI 10.7717/peerj.19316

**Table 1  Summary table of natural products found to protect against PM2.5-induced skin cell damage.**

| Plant-derived organic chemicals | Bioactive compounds | Sources of natural products | Dosage of natural products | Protective effects though specific pathway | Ref. |
|---|---|---|---|---|---|
| Phenolic compounds | Sulforaphane, a sulfur-rich phenolic compound | Broccoli, sprout | 1, 2.5, and 5 $\mu$M | • Decreased oxidative stress<br>• Increased collagen homeostasis<br>• Reduced MMP1/ CCN1<br>• Decreased melanogenesis (EDN1)<br>• Decreased inflammation (NF-$\kappa$B, IL-1$\beta$, IL-6, and TNF-$\alpha$) | *Ko et al. (2020)* |
| Phenolic compounds | Rosmarinic acid | Rosemary, sage, and peppermint | 2.5 $\mu$M 20 mg/kg/d | • Inhibit apoptosis by lowering oxidative stress<br>• Reduced allergic rhinitis by modulation of the NF-$\kappa$B pathway and Th1/Th2 balance. | *Herath et al. (2024)* and *Zhou et al. (2022)* |
| Flavonone (a type of flavonoid) | Hesperidin | Citrus fruits | 50 $\mu$M | • Reduced ROS<br>• Decreased DNA damage<br>• Inhibited MAPK, c-Jun N-terminal kinase, and p38 | *Herath et al. (2022)* and *Fernando et al. (2022)* |
| Flavonols (a type of flavonoid) | Quercetin | Berry fruits andleaves of Thunbergia laurifolia Lindl (TLE) | 25-100 $\mu$g/ml of TLE contains quercetin | • Reduced ROS<br>• Activated Nrf2<br>• Inhibited p62 signaling | *Jirabanjerdsiri et al. (2022)* |
| Flavanols (a type of flavonoid) | Catechins | Green tea extracts (GTE) contains Epigallocate-chin gallate (EGCG), a type of catechin | 750 $\mu$g/ml | • Reduced ROS<br>• Modulated genes signaling profiles associated with epidermal lipid homeostasis. | *Liao, Nie & Sun (2020)* |
| Flavanols (a type of flavonoid) | Fisetin | strawberries, apples persimmons, onions, cucumbers and grapes | 10 $\mu$M | • Reduced oxidative stress and apoptosis by inhibiting the ER stress response | *Molagoda et al. (2021)* |

Jeayeng et al. (2025), *PeerJ*, DOI 10.7717/peerj.19316

**Table 1** (*continued*)

| Plant-derived organic chemicals | Bioactive compounds | Sources of natural products | Dosage of natural products | Protective effects though specific pathway | Ref. |
|---|---|---|---|---|---|
| Tannin | Diphlorethohydroxycarmalol (DPHC),a phlorotannin | Algae | 20 μM | • Decreased endoplasmic reticulum (ER) stress<br>• Decreased mitochondrial damage<br>• Decreased autophagy | *Zhen et al. (2019)* |
| Stilbenes | Resveratrol | Grape extract | 0.1-1 μM | • Reduced ROS<br>• Decreased inflammatory cytokines | *Fernando et al. (2020)* and *Leis et al. (2022)* |
| Saponin | Ginsenoside | Ginseng | 40 μM | • Reduced ROS and Oxidative stress<br>• Decreased mitochondrial damage and apoptosis | *Li et al. (2013)* |
| Flavonoid | Flavonoid | *Cornus officinalis* extract | 200 μg/mL | • Reduced oxidative stress, DNA damage and apoptosis | *Fernando et al. (2020)* |
| Flavonoid | Isovitexin | bamboo leaves, mung beans and fenugreek seeds | 10 μM | • Inhibited ROS, enhanced the stem cell properties of keratinocytes | *Chowjarean et al. (2019)* |
| Lycium barbarum polysaccharide (LBP) | Polysaccharides | *Lycium barbarum* | 2.5 mg/ml | • Reduced cytotoxicity<br>• Decreased oxidative stress and ER stress<br>• Decreased autophagy and apoptosis | *Zhu et al. (2022)* |
| Farnesol | Sesquiterpene compound | Essential oil derived from various plants | 4 mM | • Reduced the production of pro-inflammatory cytokines<br>• Repaired<br>• PM2.5-induced injury in the epidermis and dermis | *Wu et al. (2021)* |
| Hydroxycinnamic acid | 4-O-feruloylquinic acid (FQA) | *Phellodendron amurense* | | • Reduce inflammation via PAR-2 signaling | *Choi et al. (2020)* |

pathway, where it conjugates with glutathione and undergoes biotransformation, leading to the formation of active metabolites (*Kensler et al., 2013*; *Vanduchova, Anzenbacher & Anzenbacherova, 2018*). Sulforaphane contains an isothiocyanate group that plays a critical role in protecting against electrophilic toxicities of ROS by inducing phase II biotransformation enzymes, which function as an indirect antioxidant defense system *e.g.*, catalases, glutathione S-transferases, *etc.* (*Fahey & Talalay, 1999*). This makes sulforaphane highly versatile and effective in safeguarding cells from various oxidative stresses and ROS. Growing evidence now demonstrates that the sulforaphane enhances the antioxidant capacity of animal cells, improving their ability to withstand oxidative stress (*Barton & Ollis, 1979*). A dermatological study revealed that sulforaphane supports collagen homeostasis while suppressing melanogenesis, potentially mitigating the premature aging effects induced by PM2.5 exposure (*Andres et al., 2018*). Additionally, sulforaphane disrupts melanogenic paracrine mediators, thereby inhibiting the production of melanogenic proteins and melanin in melanocytes (*Jung et al., 2013*). In keratinocytes, sulforaphane reduces the expression of NF-κB-mediated cytokines such as tumor necrosis factor α, interleukin-1β, interleukin-6 and cyclooxygenase-2 (*Fernando et al., 2019*). Sulforaphane has been reported to decrease the expression of matrix metalloproteinase-1, phospho-NF-κB, and cysteine-rich protein while significantly increasing the production of procollagen type I, indicating an anti-inflammatory effect and remodeling of the human skin fibroblasts (*Quan et al., 2012*).

## Phenolic compounds—rosmarinic acid from rosemary, sage, and peppermint

Rosmarinic acid (RA), a phenolic compound present in herbs such as rosemary, sage, and peppermint, exhibits protective effects against PM2.5-induced skin damage. This response is primarily attributed to the antioxidant, anti-inflammatory, and antiapoptotic properties of RA (*Zhou et al., 2022*). RA protects skin keratinocytes by reducing oxidative stress, as indicated by reductions in stress markers, such as lipid peroxidation, protein carbonylation, and DNA damage, which are associated with the critical macromolecules needed to maintain cellular integrity under environmental stress conditions. In addition, RA stabilizes intracellular calcium levels and mitochondrial membrane potential and prevents mitochondrial dysfunction and apoptosis, primarily by downregulating proapoptotic proteins, such as Bax and caspases, while upregulating antiapoptotic proteins, such as Bcl-2 (*Herath et al., 2024*). RA also exerts anti-inflammatory effects by modulating the NF-κB signaling pathway, which plays an important role in inflammation and immune responses. In models of allergic rhinitis exposed to PM2.5, RA treatment has been shown to reduce inflammation by decreasing proinflammatory cytokines, such as IL-4 and IL-13, and by increasing IFN-γ, thereby balancing the Th1/Th2 response (*Zhou et al., 2022*). This response suggests that the protective role of RA against PM2.5-induced damage extends beyond its antioxidant activity to include the modulation of inflammatory pathways (*Zhou et al., 2022*). Overall, RA provides multilayered protection against PM2.5-induced skin damage, highlighting its potential as a therapeutic agent that could alleviate skin problems caused by environmental pollutants.

## Protective effects of the flavonoid family compounds
### *Flavonones—Hesperidin from citrus fruit extracts*

Hesperidin (3,5,7-trihydroxyflavanone 7-rhamnoglucoside, hesperetin-7-rutinoside) is a flavonoid compound present in many citrus fruits (*Pyrzynska, 2022*). Hesperidin has also been investigated for its protective effects against PM2.5 damage in human HaCaT keratinocytes (*Herath et al., 2022*). Treating the cells with hesperidin prior to exposure to PM2.5 increased cell viability compared to untreated cells (*Fernando et al., 2022*). Hesperidin has been proposed as a potential drug because it has similar effects to those of N-acetylcysteine (NAC; a known antioxidant) because it inhibits intracellular ROS formation, cellular oxidative stress damage, mitochondrial dysfunction, autophagy, and cellular apoptosis that occurs in cells after exposure to PM2.5 (*Fernando et al., 2022*; *Herath et al., 2022*). In one study, hesperidin treatment increased keratinocyte survival from 68% (without pretreatment with hesperidin) to 82% (with pretreatment with hesperidin) after exposure to PM2.5 (*Fernando et al., 2022*).

Previous studies have shown that the target of hesperidin is an antioxidant defense system that operates *via* the Keap1-Nrf2 (Kelch-like ECH-Associating protein 1 nuclear factor erythroid 2 related factor 2) pathway, which is one of the known defense mechanisms against oxidative and electrophilic stress (*Aggarwal et al., 2020*; *Lu et al., 2016*). PM2.5 exposure results in mitochondrial damage because of alterations in mitochondrial morphology and mitochondrial respiratory chain function, resulting in mitochondrial dysfunction (*Guo et al., 2017*). Mitochondrial injury causes a mitochondrial membrane permeability transition (MMPT), resulting in mitochondrial membrane depolarization and impaired ATP production, which in turn leads to cellular apoptosis, autophagy, and damaged mitochondria (*Fernando et al., 2022*). Hesperidin treatment has cytoprotective effects on HaCaT keratinocytes through a reduction in the mitochondrial ROS formation induced by PM2.5 and the prevention of mitochondrial depolarization (*Fernando et al., 2022*; *Herath et al., 2022*).

An *in vitro* skin damage study also confirmed that hesperidin can penetrate the skin barrier to reach the dermis layer and protect the extracellular matrix from oxidative stress (*Addor, 2017*). Hesperidin shows the ability to neutralize free radicals and indirectly increase the activity of antioxidant enzymes, such as superoxide dismutase and catalase (*Estruel-Amades et al., 2019*). In addition, when used as a dietary supplement and pharmaceutical agent, hesperidin has been reported to prevent the biosynthesis of melanin and to decrease the activity of tyrosinase in both normal human melanocytes and B16F10 cells (*Katiyar et al., 2014*).

### Flavonols—Quercetin from berry fruit extracts

A study focusing on an extract from the leaves of *Thunberia laurifolia* (STLE) suggested that STLE effectively inhibits PM2.5-induced oxidative stress in keratinocytes by upregulating Nrf2 and enhancing the nuclear translocation and upregulation of p62 (*Jirabanjerdsiri et al., 2022*). STLE is a natural antioxidant mixture that contains caffeic acid, rosmarinic acid, quercetin, isoquercetin, catechin, and apigenin. Among these secondary metabolites,

quercetin is a flavonoid found in fruits and vegetables and has shown potential health-promoting effects, such as anti-inflammation, antioxidant activity, and the alleviation of allergy symptoms. Another flavonoid, rutin, is a glycoside of quercetin, and it is found in many plants, including citrus fruits, mulberries, and cranberries. Thus, flavonoid compounds have common health benefits that can be attributed to their antioxidant properties, which could reduce oxidative stress and inflammation (*Muvhulawa et al., 2022*). In dermatology, quercetin has known benefits for the skin: It can protect against skin-damaging factors, such as UV radiation, histamine, and toxic chemical compounds, and it also displays anti-allergic properties by reducing irritation and itching caused by the release of histamine (*Qi et al., 2022*).

## Flavanols—Catechins from green tea extracts

Catechins, a group of flavonoids found in green tea, are considered a potential treatment for skin diseases because of their antioxidant and anti-aging effects (*Aljuffali et al., 2022*). The bioactive components found in green tea are catechins, which include epicatechin (EC), epigallocatechin (EGC), epicatechin gallate (ECG), and epigallocatechin gallate (EGCG) (*Cardoso et al., 2020*; *Tian et al., 2021*). The various catechins differ in their chemical structures, but all show the ability to scavenge ROS, which contributes to their antioxidant, anti-inflammatory, and anticarcinogenic effects (*Wang, Chou & Hung, 2022b*). EGCG has the greatest anti-inflammatory potential due to the presence of hydroxyl groups that provide it with ROS scavenging capabilities (*Cardoso et al., 2020*). Catechins can also chelate metal ions, further enhancing their antioxidant activity. EGCG has been shown to inhibit TPA-induced DNA binding of NF-kappaB and the cAMP response element binding protein (CREB) by blocking p38 MAPK activation and causing an inactivation of the CREB transcription. This result explains how EGCG inhibits COX-2 in the mouse skin model (*Kundu & Surh, 2007*). Some studies have found that green tea extract (GTE) mitigates the damage caused by PM2.5 exposure through anti-inflammatory and antioxidant mechanisms. In addition, the restoration of impaired epidermal lipid homeostasis also plays an important role in reducing skin damage, suggesting that one effect of GTE is the restoration of PM2.5-altered gene expression in keratinocytes. *Liao, Nie & Sun (2020)* used MTT assays, morphological studies, and mass spectrometry to investigate the effect of a GTE concentration on keratinocytes growing in a three-dimensional epidermal tissue model. The polyphenol content of the GTE sample was 750 µg/mL (*Liao, Nie & Sun, 2020*). They found that GTE worked primarily by reducing the effects of PM2.5 on keratinocytes. Treatment with GTE triggered a downregulation of cytokine signaling associated with the immune system and wound healing, including the HMGCS1 and LDLR genes, which are strongly associated with the inflammatory response (*Liao, Nie & Sun, 2020*).

## Flavanols—Fisetin from various fruits and vegetables

Fisetin, a bioactive flavonoid found in various fruits and vegetables such as strawberries, apples, persimmons, onions, cucumbers, and grapes, is known for its potent antioxidant, anti-inflammatory, and antiapoptotic properties. Fisetin has been demonstrated to protect skin keratinocytes by inhibiting PM2.5-induced oxidative stress and apoptosis,

primarily through the suppression of ER stress responses (*Molagoda et al., 2021*). It effectively downregulates ER stress markers, such as glucose-regulated protein 78 (GRP78), phosphorylated eIF2α, ATF4, and CHOP, thereby reducing ER stress-induced apoptosis. Fisetin also reduces the production of ROS, lowers cytosolic calcium levels, and prevents mitochondrial dysfunction, which are critical steps for attenuating cell damage caused by PM2.5 exposure. By modulating the expression of proapoptotic proteins, such as Bax and caspases, while upregulating antiapoptotic proteins, such as Bcl-2, fisetin provides comprehensive protection against PM2.5-induced keratinocyte apoptosis (*Molagoda et al., 2021*).

## Flavonoids—from *Cornus officinalis* extract

*Cornus officinalis* extracts, and particularly its ethanol extract (EECF), have shown significant protective effects against PM2.5-induced skin damage by attenuating oxidative stress, mitochondrial dysfunction, and apoptosis in skin cells. EECF exerts its protective effect by scavenging ROS and reducing oxidative damage. EECF has been shown to attenuate lipid peroxidation, protein oxidation, and DNA damage in PM2.5-exposed keratinocytes (*Fernando et al., 2020*). In addition, EECF has been demonstrated to reduce intracellular and mitochondrial Ca2+ levels, protecting cells from mitochondrial depolarization—a key event leading to cell apoptosis. EECF has also been shown to inhibit the upregulation of proapoptotic proteins, such as Bax and cleaved caspase-3, while enhancing the expression of antiapoptotic proteins, such as Bcl-2, to further prevent PM2.5-induced apoptosis. These results suggest that a *Cornus officinalis* extract could be an effective ingredient in skin care products designed to protect the skin from environmental pollutants, such as PM2.5 (*Fernando et al., 2020*).

## Flavonoids—Isovitexin, an isomer of vitexin from bamboo leaves, mung beans, and fenugreek seeds

Isovitexin, an isomer of vitexin, is a naturally occurring flavonoid with significant antioxidant and protective properties against PM2.5-induced skin damage. Isovitexin has been shown to reduce PM2.5-induced ROS in human keratinocytes, thereby mitigating oxidative stress and preventing cell damage. Isovitexin protects skin cells by inhibiting ROS production, as demonstrated in studies in which pretreatment with isovitexin significantly reduced intracellular ROS levels in keratinocytes exposed to PM2.5. The compound exhibits potent free radical scavenging activity against DPPH, ABTS, and superoxide anion radicals, indicating a strong antioxidant capacity. In addition, isovitexin enhances stem cell properties in keratinocytes by upregulating the expression of stem cell markers, such as CD133 and β-catenin, which play roles in skin regeneration and repair (*Chowjarean et al., 2019*). These results suggest that isovitexin not only protects the skin from the damaging effects of PM2.5 but also promotes skin health by enhancing the regenerative capacity of keratinocytes.

## Tannins—Diphloroethohydroxycarmalol, a phlorotannin, from algae extracts

Diphloroethohydroxycarmalol (DPHC), a bioactive compound classified as an algal polyphenol, is derived from *Ishige okamurae*, a type of brown algae (*Bak et al., 2023*; *Diao et al., 2021*). DPHC has been previously reported to provide health benefits, including antioxidant and radioprotective effects (*Ahn et al., 2011*). Because of its health benefits and antioxidant effects, dermatological studies have been published using DPHC, which was found to reduce PM2.5-induced skin dysfunction, oxidative stress, autophagy, ER stress, mitochondrial damage, and apoptosis (*Zhen et al., 2019*). DPHC enhances cell viability by preventing oxidative stress–induced DNA damage, with a consequent inhibition of matrix metalloproteinase-1 (MMP-1), also known as collagenase enzyme, and activation of the nucleotide excision repair system (*Zhen et al., 2019*). Other findings also support a protective effect of DPHC that arises through the inhibition of lipid peroxidation and cell damage (*Zhang et al., 2021*). Treatment with DPHC in HepG2 hepatocytes cell line can reduce the production of CCAAT-enhancer-binding protein (CHOP), GRP78, and inositol-requiring enzyme 1-$\alpha$, and all three of these proteins are involved in cell apoptosis *via* the activating transcription factor 6 signaling pathway (*Zhang et al., 2021*). Notably, DPHC demonstrates effects similar to those of hesperidin in terms of mitigating PM2.5-induced mitochondrial damage arising from increased mitochondrial ROS and disruptions in the membrane permeability balance. DPHC plays a role in decreasing Bcl-associated X protein and increasing antiapoptotic Bcl-2 proteins, thereby suppressing PM2.5-induced mitochondrial damage (*Zhang et al., 2021*).

## Stilbenes—Resveratrol from grape extracts

Resveratrol (3,5,4′-trihydroxy-trans-stilbene), a natural polyphenol that can be found and extracted from grape skins and seeds (*Salehi et al., 2018*), has many health benefits, most notably antioxidant and anti-inflammatory activities and protective effects against cardiovascular disease and cancer (*Meng et al., 2020*). Resveratrol can also inhibit PM2.5-induced inflammation on the skin (*Shin et al., 2020*). Resveratrol has been reported to be safe for use on skin cells, including human keratinocytes, at a concentration of one µM or less (*Shin et al., 2020*). Pharmacokinetics studies on resveratrol have shown that 70% of resveratrol is absorbed *via* the gastrointestinal tract but then later metabolized, resulting in extremely low bioavailability (*Shaito et al., 2020*). Previous study on the larger size of particulate matter (PM) showed a reduction in intracellular ROS observed in cells pretreated with resveratrol, as resveratrol can inhibit AhR activation by PM and thus inhibit the AhR pathway. Resveratrol can also suppress the JNK/MAPK signaling pathway, which regulates COX2/PGE2 and the inflammatory cytokines induced by PM (*Shin et al., 2020*). Resveratrol can also reduce the expression of proinflammatory cytokines and IL-8, which contribute to skin inflammation and photoaging. Additionally, resveratrol treatment lowers the levels of MMP-1 and MMP-9, which are enzymes responsible for degrading the skin's extracellular matrix (*Michalak-Stoma et al., 2021*; *Shin et al., 2020*). A dosage of resveratrol of 0.1–1 µM shows the most beneficial effects (*Shin et al., 2020*).

## Saponin—Ginsenoside from ginseng extracts

Ginsenosides, a class of saponins found in ginseng, are characterized by hydroxyl groups attached to a steroid-like backbone with sugars and a distinctive four-ring structure (*Khan, Tosun & Kim, 2015*). Ginsenoside Rb1 has shown a protective effect against PM2.5-induced ROS production. Pretreatment with ginsenoside Rb1 increased cell viability from about 70% (without pretreatment) to 90% (with pretreatment). The scavenging capacity of ginsenoside Rb1 has been confirmed by a reduction in the superoxide anion signal, as measured by a xanthine oxidase system, from 2241 to 2024 signal value, and the hydroxyl radical generated by the Fenton reaction was reduced from 2844 to 2065 (*Piao et al., 2019*). Ginsenoside Rb1 protected against PM2.5-induced DNA degradation and apoptosis in HaCaT keratinocytes and normal human dermal fibroblasts (NHDF) through the proapoptotic protein Bax and the antiapoptotic proteins Bcl-2 and Mcl-1. Ginsenoside Rb1 also protected against PM2.5-induced mitochondrial damage by decreasing ROS production and reducing $Ca^{2+}$ overload, as confirmed by the measurement of ATP levels in HaCaT and NHDF cells. The cells exposed to PM2.5 alone had ATP levels of 0.795 and 0.233, respectively, while cells pretreated with ginsenoside Rb1 showed increased ATP levels of 0.94 and 0.283, respectively (*Wee, Mee Park & Chung, 2011*). Ginsenoside Rb1 at a dosage of 40 $\mu$M was effective in protecting cells *via* multiple mechanisms that led to reductions in ROS, oxidative stress, apoptosis, and mitochondrial damage (*Wee, Mee Park & Chung, 2011*). Ginseng has a wide range of health benefits, such as boosting the immune system, improving central nervous system function, alleviating stress, and scavenging free radicals (2024; *Wee, Mee Park & Chung, 2011*; *Leung & Wong, 2010*).

## Lycium barbarum polysaccharide—from *Lycium barbarum* (goji berry)

Lycium barbarum polysaccharide (LBP), extracted from the fruit of *Lycium barbarum*, has demonstrated protective effects against PM2.5-induced skin damage through its antioxidant, anti-inflammatory, and antiapoptotic activities. LBP protects skin cells, especially HaCaT keratinocytes, by inhibiting oxidative stress markers, reducing ROS levels, and restoring the activities of antioxidant enzymes, such as superoxide dismutase (SOD) and glutathione peroxidase (GPx). In addition, LBP alleviates ER stress by downregulating stress-related proteins, such as GRP78 and CHOP, which are involved in apoptotic pathways. By mitigating these stress responses, LBP reduces apoptosis, demonstrating its ability to maintain the viability of skin cells exposed to PM2.5. Furthermore, LBP modulates autophagy, a process that is often dysregulated in response to PM2.5, by affecting the expression of autophagy-related proteins, such as LC3-II and p62. This regulation helps maintain cellular homeostasis and prevents excessive autophagy, which could otherwise lead to cell death. The combined antioxidant, antiapoptotic, and autophagy-regulating properties of LBP make it a promising therapeutic agent against PM2.5-induced skin damage (*Zhu et al., 2022*).

## Farnesol—a natural benzyl semiterpene

Farnesol, a component of essential oil from various plants, such as citronella, lemongrass, tuberose, cyclamen, rose, neroli, balsam, and musk, has anti-inflammatory and antioxidant

properties and therefore provides a significant protective effect against PM2.5-induced skin damage because of its tissue-reparative properties. Farnesol exerts its protective effect by reducing the production of proinflammatory cytokines, such as IL-6 and TNF-α, which are elevated in response to PM2.5 exposure. In addition, liposomes loaded with farnesol (Lipo-Fern) can enhance the therapeutic efficacy of farnesol by improving skin penetration and reducing cytotoxicity. Studies have shown that Lipo-Fern can effectively alleviate acute and chronic inflammation, restore damaged epidermis and dermis, and promote hair follicle regeneration. This underlines its potential as a therapeutic agent against PM2.5-induced skin damage (*Wu et al., 2021*).

### 4-O-Feruloylquinic acid—from *Phellodendron amurense*

An extract (PAE) from *Phellodendron amurense*, also known as the Amur cork tree, has shown significant protective effects against PM2.5-induced skin damage through its anti-inflammatory and antioxidant properties. PAE has been demonstrated to mitigate skin damaging effects by inhibiting calcium influx and regulating signaling *via* proteinase-activated receptor-2 (PAR-2), which plays a key role in inflammatory responses. Studies have shown that PM2.5 induces an intracellular calcium influx (Ca2+) that triggers inflammatory pathways in keratinocytes. PAE significantly reduces this Ca2+ influx and thus attenuates the inflammation caused by PM2.5. In addition, PAE downregulates proinflammatory cytokines, such as IL-6, IL-8, and TNF-α, which are upregulated by PM2.5 exposure. In addition to its anti-inflammatory effects, PAE also helps maintain the integrity of the skin barrier by preventing the downregulation of proteins, such as zonula occludens-1 and occludin, both of which are crucial for tight junctions in skin cells. In addition, *Phellodendron amurense* contains 4-O-feruloylquinic acid (FQA), a compound responsible for its protective effect. FQA replicates the protective effect of PAE by inhibiting Ca2+ influx and reducing the expression of PAR-2. This suggests that *Phellodendron amurense* extract, and particularly the FQA it contains, provides significant protection against PM2.5-induced skin damage by modulating inflammatory pathways and maintaining the skin's barrier function (*Choi et al., 2020*).

## DISCUSSION

Given the potential for drug resistance associated with current medicines, alternative therapies derived from natural products have emerged as viable options for reducing drug interactions and drug resistance and mitigating the adverse effects associated with drug metabolism. However, a notable drawback of using plant extracts is that the chemical composition and antioxidant properties of the plant material can be highly variable, as the levels of secondary metabolites can depend on environmental conditions, harvest timing, and transportation and storage conditions (*Hering et al., 2021*). Therefore, methodological improvements must focus on optimizing the extraction process while considering external factors, such as temperature, light, and air, to ensure consistency in the quality and efficacy of the harvested plant materials (*Pyrzynska, 2022*).

The chemical composition and bioactivity of plant materials are highly influenced by geographical origin, soil quality, climatic conditions, seasonal variations, harvest time,

and postharvest storage conditions (*Hering et al., 2021*). These variables have a significant impact on the concentration, stability, and bioavailability of bioactive compounds, leading to variations in therapeutic efficacy. For example, flavonoids, polyphenols, and saponins exhibit region-specific variations because of differences in temperature, soil moisture, and mineral content, which affect their biosynthesis and accumulation in plant tissues. Similarly, exposure to light and oxidative stress during processing can lead to the degradation or structural modification of bioactive compounds, reducing their potency and antioxidant activity (*Biesaga, 2011*; *Pyrzynska, 2022*). An example of bioactive variability can be observed in citrus flavonoids, such as hesperidin, which have higher concentrations in the peels of citrus fruits grown in sun-exposed environments than in those grown in shaded or cooler regions (*Hu et al., 2018*). Similarly, the stability of polyphenols in plant extracts is influenced by processing methods, such as drying, grinding, solvent extraction, and heat treatment, which affect the final composition and therapeutic potential (*Cobb, 2014*). Quercetin-rich extracts, for example, vary significantly in purity and antioxidant activity depending on the choice of extraction method and solvent (*Alizadeh & Ebrahimzadeh, 2022*).

To overcome these challenges, detailed studies are needed to quantify how environmental and processing factors affect specific bioactive compounds. Future research should focus on the development of standardized cultivation, extraction, and formulation protocols to ensure consistency in the bioactive metabolite composition and therapeutic efficacy. In addition, metabolomic profiling and advanced analytical techniques, including high-performance liquid chromatography, mass spectrometry, and nuclear magnetic resonance, could be used to characterize and quantify bioactive variations in plant extracts under different environmental conditions. By identifying and controlling these variability factors, researchers can improve the reproducibility, efficacy, and safety of natural products intended for dermatological applications. The implementation of good agricultural practices (GAP), standardized extraction methods, and advanced formulation strategies is essential to minimize variability and maximize the therapeutic benefits of bioactive compounds that can protect the skin from environmental pollutants such as PM2.5.

Numerous natural compounds, including flavonoids (quercetin, hesperidin), polyphenols (resveratrol, catechins), and saponins (ginsenosides), have demonstrated significant antioxidant, anti-inflammatory, and skin-protective effects in *in vitro* and *in vivo* studies. However, their clinical application remains limited because of poor bioavailability, as well as instability and formulation challenges (*Lee et al., 2023*). Although plant-derived bioactive compounds have shown promise for disease treatment, problems such as low solubility, rapid degradation, and inefficient systemic absorption significantly restrict their therapeutic efficacy when applied topically or administered orally (*Kothapalli & Vasanthan, 2024*; *Patel et al., 2024*; *Zhao, Yang & Xie, 2019*). To overcome these limitations, recent advances in nanotechnology-based drug delivery systems have emerged as promising strategies for improving the stability, solubility, and controlled release of natural products in dermatological applications. Lipid-based nanocarriers have been particularly effective in enhancing the solubility, stability, and retention of bioactive compounds, thereby extending their therapeutic lifespan and improving transdermal penetration (*Kothapalli & Vasanthan, 2024*). Nanotechnology-based approaches, including the use of lipid-based nanoparticles,

polymeric nanoparticles, and nanoemulsions, can improve the physicochemical properties and pharmacokinetic profiles of natural bioactive compounds to ensure greater therapeutic efficiency (*Patel et al., 2024*). Various strategies, such as the use of absorption enhancers, structural transformations, and pharmaceutical technologies, have been developed to improve the oral bioavailability of poorly water-soluble flavonoids, further enhancing their systemic effects (*Zhao, Yang & Xie, 2019*).

Nanotechnology offers effective tools to enhance the bioavailability and bioactivity of phytomedicines by addressing major pharmacokinetic challenges, including poor permeation, low systemic availability, and extensive first-pass metabolism (*Gunasekaran et al., 2014*). Several nanocarrier systems, such as polymeric nanoparticles, liposomes, proliposomes, solid lipid nanoparticles, and nanoemulsions, have been developed to protect active pharmaceutical ingredients (APIs) from oxidative, hydrolytic, and environmental degradation to extend their shelf lives and enhance their therapeutic potential (*Musthaba et al., 2009*). Nanoemulsions and liposomal formulations have demonstrated superior transdermal penetration, enabling localized drug delivery while reducing irritation and ensuring sustained drug release, making them ideal candidates for treating skin conditions such as acne, eczema, psoriasis, and premature aging, all of which can be exacerbated by PM2.5 exposure (*Thakur et al., 2011*).

Transdermal drug delivery systems incorporating synergistic permeation enhancers have also been developed to optimize the absorption of both hydrophilic and lipophilic APIs and to ensure improved bioavailability while minimizing epidermal disruption. By integrating nanoencapsulation techniques with permeation enhancers, researchers have successfully increased dermal retention and systemic uptake of natural antioxidants, thus enhancing their therapeutic potential (*Schafer et al., 2023*). These strategies not only improve drug delivery efficiency but also ensure prolonged therapeutic effects, making them a promising alternative for future dermatological treatments targeting PM2.5-induced skin damage.

The skin microbiome also plays an essential role in maintaining skin homeostasis (*Kortekaas Krohn et al., 2024*). Environmental pollutants such as PM2.5 can alter microbial diversity, weaken immune defenses, and increase susceptibility to inflammatory skin diseases (*Kortekaas Krohn et al., 2024*). Investigating how bioactive compounds affect microbiome resilience and immune regulation could therefore provide new insights into their protective mechanisms against environmental skin damage (*Frasier, Fonceca & Fritts, 2024*).

Aside from formulation challenges, the source of extraction and the food matrix have a significant impact on the bioactive content and antioxidant properties of botanicals (*Cobb, 2014*). Different parts of a fruit can exhibit varying levels of bioactive compounds. For instance, the peel of citrus fruits contains higher levels of antioxidants than does the pulp, primarily because of the greater concentration of flavonoids, vitamin C, and carotenoids in the peel (*Hu et al., 2018*). Hesperidin extracted from orange peels showed moderate antioxidant activity against DPPH free radicals at 36%, while vitamin C showed a higher ability (100%) under the same conditions (*Al-Ashaal & El-Sheltawy, 2011*).

Apart from skin damage, several types of vitamins have also shown protective effects as biological agents in a variety of organ systems, such as respiratory epithelial cells, which

are protected by vitamin D (*Chatsirisupachai et al., 2024*). Despite these potential benefits, certain natural compounds, such as quercetin, are limited in their clinical usefulness because of their low solubility and bioavailability (*Alizadeh & Ebrahimzadeh, 2022*). Additionally, high doses of quercetin have been linked to renal toxicity and estrogen-dependent cancer risks, highlighting the need for more research on long-term safety and optimized dosing (*Andres et al., 2018*). While *in vitro* studies have demonstrated promising therapeutic effects, further clinical research is required to establish the optimal dosages, long-term safety, and real-world efficacy of these compounds in human skin applications. Integration of nanotechnology into the formulation of natural products represents a viable approach to overcome the inherent limitations of the bioavailability and stability of these compounds.

Addressing these existing research gaps will significantly advance the use of natural products to protect the skin from air pollution and ultimately lead to the development of innovative, science-based skin care solutions and therapeutic interventions. Future research should focus on clinical validation, safety assessments, and optimization of nanoformulated natural products for long-term use in skin care and therapeutic applications. In particular, the integration of nanoparticle-based drug delivery systems with natural bioactives presents new opportunities for more effective dermatological interventions against PM2.5-induced oxidative stress and inflammation.

## CONCLUSION

Natural products have significant potential as alternative therapeutic agents to mitigate PM2.5-induced skin damage owing to their antioxidant, anti-inflammatory, and skin barrier protective properties. Bioactive flavonoid and phenolic compounds, such as sulforaphane, hesperidin, quercetin, catechin, diphloroethohydroxycarmalol, resveratrol, and ginsenoside, have demonstrated efficacy in reducing oxidative stress, inhibiting inflammatory processes, and promoting skin regeneration, making them promising candidates as protectants for skin exposed to environmental pollutants. The mechanisms of action of these natural compounds include the elimination of ROS, inhibition of the NF-κB and MAPK signaling pathways, and enhancement of endogenous antioxidant defenses, all of which contribute to reductions in apoptosis, mitochondrial dysfunction, and cytokine-triggered inflammation in PM2.5-exposed skin cells.

## FUTURE PROSPECTIVES AND LIMITATIONS

The therapeutic effects recorded in non-English publications are often reported for traditional medicine. Limiting our search strategies to studies published in English may have resulted in a more consistent methodological approach compared to restricting eligibility criteria without narrowing the search strategy. Hence, the primary reason for restricting the inclusion criteria to English-only articles was to prevent misunderstandings arising from language diversity.

Despite their promising therapeutic effects, the clinical application of natural products remains limited because of the poor bioavailability, low stability, and limited skin penetration of these compounds. Recent advances in nanotechnology delivery systems,

such as lipid nanoparticles, nanoemulsions, and liposomes, have shown great potential to overcome these limitations by improving solubility and facilitating controlled release and transdermal absorption. These formulation strategies should pave the way for more effective dermatological applications and improved therapeutic outcomes. Future research should focus on optimizing formulations, standardizing extraction methods, and conducting rigorous clinical trials to validate the long-term efficacy and safety of natural product–based interventions. In addition, studying the interaction between natural products and the skin microbiome could provide new insights into the mechanisms by which they protect skin exposed to polluted environments. The integration of nanotechnologically active ingredients with natural bioactivities is a promising avenue for the development of innovative, science-based skin care solutions. Addressing these existing research gaps will significantly enhance the use of natural active compounds in protecting the skin from PM2.5-induced oxidative stress, inflammation, and barrier disruption, ultimately contributing to safer and more effective dermatological therapies.

## ACKNOWLEDGEMENTS

We thank Monica Madore from Scribendi for editing a draft of this manuscript.

### Funding

This research project is supported by Chulabhorn Royal Academy (Fundamental Fund by National Science Research and Innovation Fund (NSRF): fiscal year 2024) (FRB670024/0240 Project code 198474), Chulabhorn Royal Academy (Project code E2567/106) and the Princess Srisavangavadhana Faculty of Medicine. The funders had no role in study design, data collection and analysis, decision to publish, or preparation of the manuscript.

### Grant Disclosures

The following grant information was disclosed by the authors:
Chulabhorn Royal Academy (Fundamental Fund by National Science Research and Innovation Fund (NSRF)(FRB670024/0240 Project code 198474).
Chulabhorn Royal Academy (Project code E2567/106).
The Princess Srisavangavadhana Faculty of Medicine.

### Competing Interests

The authors declare there are no competing interests.

### Author Contributions

- Saowanee Jeayeng conceived and designed the experiments, performed the experiments, analyzed the data, prepared figures and/or tables, authored or reviewed drafts of the article, and approved the final draft.

- Jaturon Kwanthongdee conceived and designed the experiments, performed the experiments, analyzed the data, prepared figures and/or tables, authored or reviewed drafts of the article, and approved the final draft.
- Ratima Jittreeprasert performed the experiments, analyzed the data, prepared figures and/or tables, authored or reviewed drafts of the article, and approved the final draft.
- Kankanich Runganantchai performed the experiments, analyzed the data, prepared figures and/or tables, authored or reviewed drafts of the article, and approved the final draft.
- Kalayaporn Naksavasdi performed the experiments, analyzed the data, prepared figures and/or tables, authored or reviewed drafts of the article, and approved the final draft.
- Rosarin Rirkkrai performed the experiments, analyzed the data, prepared figures and/or tables, authored or reviewed drafts of the article, and approved the final draft.
- Varisara Wongcharoenthavorn performed the experiments, analyzed the data, prepared figures and/or tables, authored or reviewed drafts of the article, and approved the final draft.
- Wiriya Mahikul conceived and designed the experiments, performed the experiments, analyzed the data, prepared figures and/or tables, authored or reviewed drafts of the article, and approved the final draft.
- Anyamanee Chatsirisupachai conceived and designed the experiments, performed the experiments, analyzed the data, prepared figures and/or tables, authored or reviewed drafts of the article, and approved the final draft.

### Data Availability

This is a literature review.

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
