# Peer review of "Natural products as promising therapeutics for fine particulate matter–induced skin damage: a review of pre-clinical studies on skin inflammation and barrier dysfunction"

_PeerJ, doi:10.7717/peerj.19316_

## Round 0.1 · original submission · Major Revisions

Both reviewers raise important questions, please address them carefully.

Reviewer 1 ·

Basic reporting

Manuscript ID; PeerJ_110893
Title: Natural products as promising therapeutics for fine particulate matter-induced skin damage: review of pre-clinical studies on skin inflammation and barrier dysfunction

Although the topic is of interest to the scientific community, this paper should be improved before being considered for publication in any academic journal. Authors should reconsider the main objective of the paper according to its content. They should try to synthesize and emphasize the main findings of the review contents and avoid long sentences. Additionally, the paper's structure needs improvement for better readability and coherence. In addition, the conclusion is not well-written and fails to summarize the findings and highlight their significance effectively.

Experimental design

Line 141 The importance of natural products to protect against PM2.5-induced skin damage
- The bioavailability of these natural compounds and their stability in topical formulations or oral supplements is not addressed. Research into improving delivery systems (e.g., nanoparticles or liposomes) for enhanced skin penetration and sustained activity is lacking. Plesee address it.
- PM-induced skin conditions, such as premature aging, acne, or skin cancer, is not discussed.
- Addressing these gaps could advance the understanding and application of natural products for protecting skin from air pollution.

Validity of the findings

Discussion; The discussion acknowledges variability in the chemical composition of plant materials due to environmental and processing factors, but more detailed research is needed to quantify how these variables affect specific bioactive compounds.

Additional comments

1. It's a nice try; But, the authors should go for the English edition. Some of the discussions are repetitive. Try to solve this problem. Otherwise, it is boring to read the manuscript.
2. Abstract: The authors should revise the abstract; it is too general. Moreover, it could be further developed, as the article has a lot of interesting data. An informative and representative conclusion should be added to the abstract.
3. Keywords: It is crucial to revise the keywords, ensuring they are spelled correctly and avoid general, abbreviation, and plural terms and multiple concepts (avoid, for example, 'and', 'of'). This will help to maintain the precision and clarity of the manuscript.
e.g. particulate matter 2.5 (PM2.5) (redundancy)
4. You must provide all the figures in high resolution and make the labels and legends more legible.
5. Conclusion: The findings could be further developed; the article contains a lot of interesting data.

·

Basic reporting

Language and Structure: The language is clear, adhering to the standards of academic writing. Introduction and Background: The introduction provides a detailed overview of the potential harm of PM2.5 to the skin and the potential of natural products as therapeutic agents, offering sufficient background information and clarifying the motivation and objectives of the study. Literature Review: The literature cited is extensive and highly relevant, covering significant studies from 1999 to 2024, providing a comprehensive overview of the field. However, some sections (e.g., lines 82-96) lack references, which need to be supplemented to enhance the rigor of the arguments. Scope and Innovation: The study focuses on the protective effects of natural products against PM2.5-induced skin damage, a topic with significant practical relevance and innovation. While research on natural products is abundant, systematic reviews specifically addressing PM2.5-induced skin damage are relatively rare, making this article valuable in filling a gap in the literature.

Experimental design

Objectives and Methods: The research objectives are clearly defined, aiming to explore the mechanisms of natural products in alleviating PM2.5-induced skin damage and providing references for future clinical applications. The methods are sound, involving a systematic search of PubMed and Google Scholar databases to select 41 relevant articles for review, ensuring comprehensiveness and objectivity. Scope and Limitations: The study scope is limited to original English-language research articles, which helps focus on basic research findings. However, this approach may omit important non-English or clinical studies, a limitation that should be noted in the discussion.

Validity of the findings

Relevance of Conclusions to Research Questions: The conclusions effectively address the research questions, clearly highlighting the potential of natural products in antioxidant, anti-inflammatory, and skin barrier protection, while emphasizing the need for further research. The conclusions are closely linked to the preceding research content without overstepping the scope or making undue inferences. Sufficiency of Argumentation: The article supports its arguments with extensive literature, detailing the mechanisms and experimental results of various natural products (e.g., sulforaphane, hesperidin, quercetin). However, some studies on specific compounds may suffer from insufficient sample sizes, imperfect experimental designs, and the use of single-cell arguments, which may affect the generalizability of the conclusions.

Additional comments

1. The structure of the article is composed of an introduction, methodology, and discussion, but lacks a conclusion section. 2. The sentence structure is disorganized, lacking a clear connection between the subject and the predicate. For example, in line 161, "Phenolic compounds, presented in a variety of foods including fruits, vegetables and beverages, which can exhibit antioxidan……" It is suggested to change it to "Phenolic compounds, which are present in a variety of foods including fruits, vegetables, and beverages, exhibit antioxidant..." 3. The sentence structure in the original text is rather complex, as seen in line 449, which includes multiple parallel elements, making the sentence lengthy and difficult to understand. Additionally, the phrase "Given the potential for drug resistance associated with medicine treatments" is questionable, as the issue has not been previously mentioned. Moreover, drug resistance may not be the most significant issue when it comes to using natural products as alternatives to chemical drugs for treating skin diseases.

---

## Round 0.2 · accepted · Accept

The manuscript can be accepted now.

Reviewer 1 ·

Basic reporting

This revised version is suitable for publication in PeerJ.

Experimental design

-

Validity of the findings

-

Additional comments

-